# RETRACTED: HOXA11-AS1 Promotes PD-L1-Mediated Immune Escape and Metastasis of Hypopharyngeal Carcinoma by Facilitating PTBP1 and FOSL1 Association

**DOI:** 10.3390/cancers14153694

**Published:** 2022-07-29

**Authors:** Zheng Zhou, Qian Liu, Gehou Zhang, Diab Mohammed, Sani Amadou, Guolin Tan, Xiaowei Zhang

**Affiliations:** 1Department of Otolaryngology Head & Neck, Third Xiangya Hospital, Changsha 410013, China; zz198911252022@163.com (Z.Z.); windyliusong@126.com (Q.L.); z1713866573@163.com (G.Z.); guolintan@csu.edu.cn (G.T.); 2Department of Otolaryngology Head & Neck, Xiangya Hospital, Changsha 410008, China; Diabsam@hotmail.com; 3Department of ENT, Reference Hospital, Maradi 12481, Niger; hamedsani86@gmail.com

**Keywords:** hypopharyngeal squamous cell carcinoma, HOXA11-AS1, PD-L1, FOSL1, PTBP1, immune escape

## Abstract

**Simple Summary:**

The metastasis of hypopharyngeal squamous cell carcinoma (HSCC) is the main reason for the poor prognosis of patients. Increasing studies have shown that abnormally expressed lncRNAs play crucial roles in HSCC, providing new perspectives for exploring cancer pathogenesis and matastasis. The expressions of HOXA11-AS1 and PD-L1 were found to be closely related to the overall survival of HSCC patients. Subsequently, the potential target genes, namely PBTP1 and FOSL1, were identified by expression correlation analysis. Finally, HOXA11-AS1/FOSL1/PTBP1/PD-L1 axis was identified to be a novel pathway provided a feasible preliminary basis for the future application of immunotherapy or targeted therapies in HSCC.

**Abstract:**

Background: The metastatic characteristics of hypopharyngeal squamous cell carcinoma (HSCC) lead to many diagnostic and therapeutic challenges, while functional long non-coding RNAs (lncRNAs) can provide effective strategies for its diagnosis and treatment. Methods: RT-qPCR, Western blot, immunohistochemistry, and an immunofluorescence assay were used to detect the related gene expression. Flow cytometry was used to measure the percentage of CD8^+^ and CD4^+^ T cells. CCK-8 and transwell assays were performed to analyze the role of HOXA11-AS1. The targeted relationship of the FOSL1/PD-L1 promoter was measured by ChIP and dual-luciferase reporter assays. RNA pulldown and RIP assays were used to measure the interaction between HOXA11-AS1, FOSL1, and PTBP1. A tumor xenograft study was used to analyze HOXA11-AS1 function in vivo. Results: HOXA11-AS1, PD-L1, and FOSL1 were upregulated in HSCC, and HOXA11-AS1 positively correlated with PD-L1. HOXA11-AS1 knockdown upregulated CD8^+^ T cells through an increase in IFN-γ concentration while decreasing the proliferation, migration, and invasion of HSCC cells. FOSL1 bound the PD-L1 promoter, increasing gene expression. HOXA11-AS1 enhanced the stability of FOSL1 mRNA by binding to PTBP1. HOXA11-AS1 or PTBP1 overexpression increased FOSL1 and PD-L1 expression. PD-L1 knockdown arrested the inhibiting function of HOXA11-AS1 overexpression on CD8^+^ T cell content. HOXA11-AS1 knockdown inhibited immune escape and metastasis through PD-L1 regulation by downregulating FOSL1 in vivo. Conclusion: HOXA11-AS1 promoted PD-L1 expression by upregulating FOSL1 levels through PTBP1, thereby facilitating immune escape, proliferation, and metastasis of HSCC cells.

## 1. Introduction

Hypopharyngeal squamous cell carcinoma (HSCC) is one of the most frustrating and aggressive malignancies in the hypopharynx, accounting for 3–5% of all head-and-neck tumors [1]. There were 84,252 cases of HSCC and 38,599 deaths in 2020 worldwide [2]. Patients with HSCC are often in an advanced stage at the time of diagnosis, and the delayed diagnosis is due to the absence of initial symptoms. Its metastatic characteristics, such as submucosal spread and cervical lymph node metastasis, and high incidence of recurrence seriously affect five-year survival rates, which are less than 40% in stage III and IV patients, who show no encouraging improvement with contemporary treatments [3,4,5]. Therefore, discovering molecular targets for HSCC diagnosis and treatment remains key to achieving clinical improvements.

Programmed death-ligand 1 (PD-L1, also named CD274 or B7-H1), one of the ligands of PD-1, acts as a coinhibitory molecule in cancer development and is expressed on the surface of T lymphocytes, facilitating immune escape [6,7]. In cancer, the activity of the PD-1/PD-L1 pathway is responsible for T cell activation, proliferation, and cytotoxic secretion, leading to the degeneration of anti-tumor immune responses [8]. Positive PD-L1 staining was found in 58% of cancers, and no survival benefit was observed from PD-L1 levels in tumor cells [9]. Growing evidence has shown that PD-L1 upregulation is involved in tumor aggressiveness, being closely related to poor prognosis [10]. For example, Wintterle et al. suggested that PD-L1 modulated CD4^+^ and CD8^+^ T cell levels by increasing cytokine (such as IFN-γ) concentrations, thus showing an immunomodulatory function in glioma cells [11]. Similarly, the high expression of PD-L1 facilitated HSCC cell escape from recognition by the host immune system and accelerated tumor metastasis [12,13], indicating that PD-L1 inhibition may have positive effects on HSCC treatment. Indeed, the measurement of PD-L1 scoring was considered as a method to evaluate the effect of (neo)adjuvant dual anti-PD-1 immune-checkpoint blockade in recurrent, resectable squamous cell carcinoma of the head and neck [14]. However, the specific function of PD-L1 in HSCC requires further investigation.

The transcription factor FOSL1 is involved in promoting metastasis in a variety of cancers, such as breast cancer [15], clear cell renal cell carcinoma [16], and bladder cancer [17]. Although the role of FOSL1 in HSCC remains largely unknown, the TCGA database predicted high FOSL1 expression in head-and-neck cancer, with microarray assays also indicating its overexpression in HSCC. More importantly, JASPAR prediction identified the FOSL1 binding sequence on the PD-L1 promoter. Therefore, we suspected that exploring their interaction may shed light on HSCC immune escape and metastasis. Increasing studies have shown that abnormally expressed long non-coding RNAs (lncRNAs) play roles as tumor promoters or suppressors, providing new perspectives for exploring cancer pathogenesis [18,19]. The lncRNA UCA1 had been reported to be highly expressed in HSCC, acting as a pro-metastatic gene and a tumor promoter in vitro [20]. This study was conceived to elucidate the molecular mechanism of homeobox A11 antisense RNA1 (HOXA11-AS1) in HSCC. HOXA11-AS1 was suggested to be an oncogene in prostate cancer [21], non-small cell lung cancer [22], and HSCC. Xu et.al suggested that HOXA11-AS1 knockdown repressed the proliferative and migration capacity of FaDu cells [23]. Our microarray analysis indicated that HOXA11-AS1 was upregulated in HSCC. Remarkably, Starbase predicted binding sequences between HOXA11-AS1 or FOSL1 and polypyrimidine tract binding protein 1 (PTBP1), which belongs to the family of heterogeneous nuclear ribonucleoproteins. PTBP1 acted as a regulator of tumorigenesis, proliferation, metastasis, and apoptosis in multiple cancers [24]. The function of PTBP1 in cancers is regulated by a variety of molecules, such as microRNAs, lncRNAs, and RNA-binding proteins. For example, the lncRNA LUCAT1/PTBP1 axis was demonstrated to promote tumor progression by regulating the selective splicing and stability of mRNA [25]. However, no studies have discussed the mechanism and role of PTBP1 in HSCC. Here, we speculated that HOXA11-AS1 promoted PTBP1 and FOSL1 association and participated in HSCC metastasis.

In this study, we constructed a network linked to the immune escapee and metastasis of HSCC by a series of analytic processes. We first evaluated the expression of HOXA11-AS1 and PD-L1 in HSCC tissues and cells and conducted survival analysis of HSCC patients. Subsequently, the potential target genes, namely PBTP1 and FOSL1, were identified by expression correlation analysis. Finally, a series of molecular biology methods were used to explore the possible upstream regulatory mechanism and downstream regulatory pathway, and also to verify the biological functions of this novel pathway. The HOXA11-AS1/FOSL1/PTBP1/PD-L1 axis provided a feasible preliminary basis for the future application of immunotherapy or targeted therapies in HSCC.

## 2. Materials and Methods

### 2.1. Patients

A total of 40 samples of HSCC and matched peri-carcinomatous specimens were obtained from patients with HSCC at the Third Xiangya Hospital of Central South University from January 2016 to June 2020. All samples were stored at −80 °C until analysis. The study was approved by the Ethics Committee of the Third Xiangya Hospital of Central South University, and all subjects signed written informed consent (IRB code: 2020-S031). The criteria for inclusion: (1) pathological diagnosis of HSCC in the resected specimens; (2) no history of radiotherapy or chemotherapy; (3) age of 18 years or older; (4) the patient’s written informed consent. Exclusion criteria: (1) history of surgical resection of any part of the upper gastrointestinal tract; (2) younger than 18 years of age; (3) current pregnancy. 

### 2.2. Cell Culture and Transfection 

Human pharyngeal carcinoma cells (FaDu and Detroit 562) and human nasopharyngeal epithelial cells (NP69) were obtained from the American Type Culture Collection (Manassas, VA, USA). Cells were cultured in Dulbecco’s modified Eagle medium (Thermo Fisher Scientific, Inc. Waltham, MA, USA) supplemented with 10% fetal calf serum (Invitrogen, Carlsbad, CA, USA), 100 U/mL penicillin, and 100 μg/mL streptomycin (Invitrogen) at 37 °C in a humidified incubator under 5% CO_2_.

Short hairpin RNAs (shRNAs) knocking down HOXA11-AS1, PD-L1, and PTBP1 and lentiviral plasmids overexpressing HOXA11-AS1 and PTBP1 were synthesized by RiboBio (Guangzhou, China). All plasmids and their combinations were transduced into FaDu or Detroit 562 cells using Lipofectamine 3000 (Invitrogen).

### 2.3. RNA Extraction and Quantitative Real-Time PCR (qRT-PCR) Assay

Total RNA from tissues or cells was extracted using TRIzol reagent (Invitrogen), and its concentration was measured using a NanoDrop Spectrophotometer (Thermo). The RevertAid First Strand cDNA Synthesis kit (Thermo) was used for reverse transcription. An ABI qRT-PCR 7900 system (Thermo) was employed to perform PCR reactions. Gene expression was normalized to GAPDH. Primers were synthesized by RiboBio and were: 5′-CGGCTAACAAGGAGATTTGG-3′ (sense) and 5′-AGGCTCAGGGATGGTAGTCC-3′ (antisense) for HOXA11-AS; 5′-GTGGCATCCAAGATACAAACTCAA-3′ (sense) and 5′-TCCTTCCTCTTGTCACGCTCA-3′ (antisense) for PD-L1; 5′-CAGTGGATGGTACAGCCTCA-3′ (sense) and 5′-CTGCAGCCCAGATTTCTCAT-3′ (antisense) for FOSL1; 5′-CTCAAGGCGTTCCTTCTGCTTC-3′ (sense) and 5′-GGAGGAGTGGGTGTCGCTGT-3′ (antisense) for GAPDH.

### 2.4. Western Blot

Protein extraction was carried out with a lysis buffer (Beyotime Institute of Biotechnology, Haimen, China). Proteins were quantified by an acid-based Protein Assay Kit (Thermo), followed by separation with 10% sodium dodecyl-sulfate polyacrylamide gel electrophoresis (SDS-PAGE) and transfer to polyvinylidene difluoride membranes (Millipore, Bedford, MA, USA). Afterward, 5% skim milk was used to block (2 h) the non-specific binding sites of membranes. Membranes were subsequently incubated with indicated primary antibodies overnight against PD-L1 (1:1000, ab213524, Abcam, Cambridge, UK), FOSL1 (1:1000, ab252421, Abcam), or PTBP1 (1:5000, ab134950, Abcam), followed by incubation with horseradish peroxidase-conjugated secondary antibody (1:5000, ab205718, Abcam). Membranes were washed with PBS three times, and signals were visualized by an ECL reagent (Millipore, Bedford, MA, USA). Protein expression was analyzed using ImageJ software 6.0 (National Institutes of Health, Bethesda, MD, USA). GAPDH (1:2000, ab8245, Abcam) and β-actin (1:5000, ab6276, Abcam) acted as internal controls.

### 2.5. Immunohistochemistry (IHC)

HSCC tumor tissues were fixed in 4% paraformaldehyde for 24 h, then routinely embedded in paraffin and sectioned into 4 µm sections. Sections were deparaffinized, hydrated, and immersed in blocking solution (prepared using Triton X-100) in the dark for 1 h. After incubating with anti-PD-L1 (1:500, ab228415, Abcam) overnight, sections were incubated with horseradish peroxidase-labeled secondary antibody (1:5000, ab205718, Abcam) for 30 min. After developing with a diaminobenzidine substrate, sections were counterstained with hematoxylin. Images were visualized using a Nikon ECLIPSE Ti microscope system and processed with Nikon software (*Version* 5.02).

### 2.6. CCK-8 Assay

Cells were seeded into culture plates and incubated for indicated times, followed by treatment with 10 µL CCK-8 solution. Absorbance was detected at 450 nm by a Multiskan^TM^ GO microplate spectrophotometer (Thermo).

### 2.7. Colony Formation Assay

After trypsinization, HSCC cells (1 × 10^5^) and PBMCs were cocultured in 6-well plates for 2 weeks under a humidified atmosphere. Colonies were fixed with 4% paraformaldehyde and then stained with 0.1% crystal violet. A microscope was used to calculate the number of colonies.

### 2.8. Wound Healing Assay

Briefly, treated cells in 6-well plates were cultured at 37 °C until 100% confluence, followed by scratching a straight wound on the surface of the cell layer using a sterile pipette. Next, debris on the cell surface was removed by washing with PBS twice, and cells were incubated for 24 h. We obtained photographs of migrating cells 0 h and 24 h after scratching using a microscope.

### 2.9. Transwell Assay

Cell invasion capability was measured by Transwell chambers with an 8 μm pore size (Corning, Tewksbury, MA, USA) coated with Matrigel. Briefly, 1 × 10^5^ cells in 100 μL of serum-free RPMI 1640 medium were placed into the upper chamber, and 500 μL of medium containing 10% FBS was added into the lower chamber. Cells were dyed in 0.5% crystal violet after 48 h culture at 37 °C with 5% CO_2_. Cells remaining in the upper chamber were removed. Following washing with PBS and drying, images were taken under a microscope (Olympus, Tokyo, Japan).

### 2.10. Co-Culture of Human PBMCs and HSCC Cells

FaDu and Detroit 562 cells with or without PD-L1 antibody (2 µg/mL, Abcam) treatment were selected for co-culture experiments. Human peripheral blood mononuclear cells (PBMCs) were isolated by Ficoll-Hypaque gradient centrifugation. For co-culture, FaDu and Detroit 562 cells were plated into 24-well plates in Modified Eagle Medium (DMEM) with a density of 1 × 105 cells/well, and PBMCs were placed in culture insert (0.4 µm pore size, Corning, New York, NY, USA) in DMEM at a concentration of 4 × 105 cells/well. 

### 2.11. Flow Cytometry

Co-cultured cells were stained with anti-CD3 (317320, BioLegend, San Diego, CA, USA), anti-CD4 (357402, BioLegend), and anti-CD8 (344702, BioLegend), and then assessed using a BD FACSAria III flow cytometer (BD Biosciences, San Jose, CA, USA). Data were analyzed with FlowJo V10 software (TreeStar Inc., Ashland, OR, USA).

### 2.12. Enzyme-Linked Immunosorbent Assay (ELISA)

Interferon-γ (IFN-γ) concentration in cell-free supernatant from co-cultured cells was assessed by a human ELISA kit according to the manufacturer’s instructions (Multisciences, Hangzhou, China).

### 2.13. Immunofluorescence Assay

Transfected cells were fixed with 4% formaldehyde for 20 min at room temperature. After 1 h of incubation with non-fat dry milk diluted in 5% Tris-buffered saline with Tween-20 (pH 8.3), cells were incubated with the primary antibody PD-L1 (1:200, ab228415, Abcam) overnight at 4 °C, and then, the corresponding secondary antibody (1:1000, ab150077, Abcam) at 37 °C for 1 h. Next, DAPI was used to stain cell nuclei, and a confocal laser scanning microscope was used to analyze immunofluorescence images.

### 2.14. Fractionation of Nuclear/Cytoplasmic RNA

Nuclear/cytoplasmic fractionation was performed using a PARIS Kit (Life Technologies, Carlsbad, CA, USA). Briefly, cells were washed with PBS, followed by centrifugation and lysis in 1 mL of lysis buffer. Cytoplasmic RNA was subsequently obtained from the supernatant, and the remaining nuclear pellet was washed three times with hypotonic lysis buffer to extract nuclear RNA.

### 2.15. Chromatin Immunoprecipitation (ChIP) Assay

The two binding sites (BS1: TGTGTCAT; BS2: AAATCACTGAGCAGCAAGCTGA) between FOSL1 and PD-L1 were predicted using the JASPAR website (http://jaspar.genereg.net/ accessed on 1 July 2021). A kit from Millipore was used to perform the ChIP assay. Briefly, cells were cross-linked with 1% formaldehyde for 10 min, then lysed and sonicated to obtain chromatin fragments of 500 bp average size. Subsequently, 1% of the supernatant was collected to serve as input control. Chromatin diluted in the ChIP solution was immunoprecipitated with a DNMT3A antibody or IgG at 4 °C overnight with rotation. After reversing the cross-links, immune complexes were purified and analyzed by PCR.

### 2.16. Dual-Luciferase Reporter Assay

The PD-L1 promoter fragment containing two binding sites (wild-type BS1 and mutant BS2, or mutant BS1 and wild-type BS2) was subcloned into a pGL3-basic vector (Promega, Madison, WI, USA), obtaining recombinant plasmids mut-BS2 and mut-BS1. FOSL1 cDNA was PCR-amplified and inserted into the pcDNA3.1 vector (Promega), with an empty vector (EV) used as control. In addition, 293T cells were co-transfected with a pGL3 luciferase construct (mutant BS1 or mutant BS2) for FOSL1 expression or empty plasmids. The luciferase signal of indicated transfected cells was estimated using a Dual-Luciferase Reporter Assay System (Promega) 48 h after transfection.

### 2.17. mRNA Stability Analysis

HOXA11-AS1- or PTBP1-silenced cells were treated with 5 μg/mL actinomycin D (MedChemExpress, Monmouth Junction, NJ, USA). Total RNA was extracted at indicated times, and HOXA11-AS1 or FOSL1 mRNA was measured and normalized to GAPDH levels using RT-qPCR.

### 2.18. RNA Pulldown

Biotin-labeled HOXA11-AS1 and FOSL1 mRNA or their antisense RNA were transcribed with the Biotin RNA Labeling Mix and T7/SP6 RNA polymerase (Roche Diagnostics, Indianapolis, IN, USA), followed by RNase-free DNase I (Roche) treatment. After purification with the RNeasy Mini Kit (Roche), biotin-labeled RNAs were mixed with extracted FaDu and Detroit 562 cell nuclear proteins and then incubated with streptavidin agarose beads at room temperature for 1 h. Protein bands were visualized by silver staining followed by Western blot.

### 2.19. RNA Immunoprecipitation (RIP) Assay

A RIP assay was conducted using a Magna RIP^TM^ RNA-Binding Protein Immunoprecipitation Kit (Millipore). Cells were lysed with RIP lysis buffer and then immunoprecipitated with immunoglobulin G antibody (anti-IgG) and argonaute 2 antibody (anti-Ago2) coated on magnetic beads overnight, followed by a PBS wash. Part of the cell lysate (“Input”) was used as a negative control. The co-precipitated RNA was extracted using TRIzol^TM^ (Ambion Inc., ThermoFisher Scientific, Waltham, MA, USA), and RT-qPCR was used to analyze the purified RNA.

### 2.20. In Vivo Xenograft Experiments

The 48 NOD-SCID mice (male, 18–22 g, 6 weeks of age) used in this study were purchased from Hunan slake Jingda Co., Ltd. (Changsha, China). FaDu or Detroit 562 cells stably expressing shHOXA11-AS1-1, shHOXA11-AS1-2, or shNC were injected subcutaneously into the right flank of mice (*n* = 4) to perform the tumorigenesis assay. Next, each xenograft tumor model was assigned into 2 groups (*n* = 4) and was injected intraperitoneally with PBS or PBMC for killing the xenograft effectively. Human immunocyte mixtures containing PBMCs (4 × 106/per mouse) and activated T cells (1 × 106/per mouse) or PBS were injected into the mice when the tumor volume was more than 100 mm^3^ by tail intravenous injection once a week for three times. Tumor volumes were calculated every 5 days by the equation: volume (mm^3^) = 0.5 × (Width)^2^ × (Length). At 30 days after injection, mice were euthanized to obtain tumors. Tumor tissue sections were stained with hematoxylin and eosin (HE) or ki67 to analyze the pathology or proliferation of tumor tissue. IHC staining was performed to measure PD-L1 levels. Tumor regression rate = (tumor volume in NOD-SCID mice after PBMC treatment/tumor volume in NOD-SCID mice after PBS treatment) × 100%.

For the metastasis assay, FaDu or Detroit 562 cells stably expressing shHOXA11-AS1-1, shHOXA11-AS1-2, and shNC were injected into the mouse tail vein (*n* = 4). Four weeks after injection, mice were sacrificed, and lung lobes were harvested and placed in 10% neutral-buffered formalin fixative overnight and embedded in paraffin. Then, sections were stained with HE for pathological assessment. Metastasis foci within mice lungs and visible lung metastatic nodules were observed and counted using a light microscope (Olympus Corporation, Tokyo, Japan). All animal experiments were approved by the Animal Ethics Committee of the Third Xiangya Hospital of Central South University (IRB code: 2020-S031).

### 2.21. Statistical Analysis

Data are provided as mean and standard deviation (SD). The Student’s *t*-test was used to compare the difference between two groups for continuous variables. One-way analysis of variance (ANOVA) followed by Tukey’s post hoc test was used for multiple comparisons. Kaplan–Meier with log-rank methods was used in survival analysis, and Pearson’s correlation was evaluated. All analyses were performed using GraphPad Prism 6 (GraphPad Software,Version 8.2.0 Inc., La Jolla, CA, USA); *p* < 0.05 was considered statistically significant.

## 3. Results

### 3.1. HOXA11-AS1 and PD-L1 Were Highly Expressed in HSCC, and HOXA11-AS1 Positively Correlated with PD-L1

This study evaluated HOXA11-AS1 and PD-L1 levels in HSCC and peri-carcinomatous tissues. As shown in Figure 1A,B, HOXA11-AS1 and PD-L1 were highly expressed in HSCC tissues. HSCC patients with lower levels of HOXA11-AS1 or PD-L1 had increased survival rates compared with those with higher levels of HOXA11-AS1 or PD-L1 (Figure 1C,D and Appendix A). Then, a positive correlation between HOXA11-AS1 and PD-L1 was confirmed (Figure 1E). To further verify the upregulation of PD-L1 in HSCC tissues, IHC staining was adopted to measure PD-L1 levels in tumor tissues. As shown in Figure 1F, PD-L1 was overexpressed. Moreover, we detected increased expression of HOXA11-AS1 and PD-L1 in HSCC cell lines Detroit 562 and FaDu, compared to NP69, a normal human nasopharyngeal epithelial cell line. (Figure 1G,H).

### 3.2. HOXA11-AS1 Knockdown Suppressed PD-L1 Expression and Immune Escape, Proliferation, and Metastasis of HSCC Cells

To explore the cellular role of HOXA11-AS1, two independent plasmids were used to deplete HOXA11-AS1, with increased specificity and helping eliminate off-target effects from the use of a single plasmid in FaDu and Detroit 562 cell lines. HOXA11-AS1 was efficiently silenced by transfecting with shHOXA11-AS1-1 and shHOXA11-AS1-2 (Figure 2A). Since HOXA11-AS1 was positively associated with PD-L1, we measured PD-L1 levels after silencing HOXA11-AS1 and observed its downregulation (Figure 2B,C). The dysregulation of PD-L1, which could regulate the proliferation and cytotoxicity of T cells, had been reported to be a cogent mechanism for potentially immunogenic tumors to escape from host immune responses [10,26]. Thus, HOXA11-AS1-silenced cells were co-cultured with PBMCs to evaluate whether the reduction in PD-L1 induced by HOXA11-AS1 knockdown could affect the concentration of CD8^+^ and CD4^+^ T cells in vitro. As shown in Figure 2D–E, the knockdown of HOXA11-AS1 or anti-PD-L1 treatment could increase CD8^+^ T cell percentages while decreasing CD4^+^ T cells in PBMCs. As expected, HOXA11-AS1 knockdown and PD-L1 antibody addition together further upregulated CD8^+^ T cell contents while downregulating CD4^+^ T cell contents, suggesting the activation of T lymphocytes. Furthermore, the knockdown of HOXA11-AS1 or anti-PD-L1 treatment increased the concentration of IFN-γ, and HOXA11-AS1 silencing and PD-L1 antibody addition further promoted IFN-γ secretion by PBMCs (Figure 2F), further supporting our conclusion that HOXA11-AS1 facilitates immune escape in HSCC cells. Next, we analyzed the effects of HOXA11-AS1 knockdown on the proliferation and metastasis of HSCC cells using CCK-8 and colony formation assays. The results from the CCK-8 assay suggested that cell viability was reduced after HOXA11-AS1 silencing (Figure 2G). Using a colony formation assay, we demonstrated that HOXA11-AS1 silencing inhibited the colony formation (Figure 2H). Similarly, the capabilities of cell migration and invasion were evaluated by wound healing and transwell assay, respectively. The results indicated that the capabilities of cell migration (Figure 2I) and invasion (Figure 2J) were repressed after HOXA11-AS1 knockdown. Taken together, HOXA11-AS1 promoted cell proliferation, metastasis, and immune escape by regulating PD-L1 in vitro.

### 3.3. FOSL1 was Highly Expressed in HSCC Cells and Positively Regulated PD-L1 Expression by Binding the PD-L1 Promoter

The TCGA database predicted that FOSL1 was significantly expressed in head-and-neck cancers. Therefore, we explored its expression and potential pathway in HSCC cells. Using qRT-PCR, we found that FOSL1 was significantly elevated in HSCC tumor samples (Figure 3A). Moreover, we found that HSCC patients with high expression of FOSL1 exhibited a worse survival rate than those with low FOSL1 expression (Figure 3B). FOSL1 was also overexpressed in FaDu and Detroit 562 cells (Figure 3C). Then, FOSL1 was successfully depleted by two plasmids, inducing the downregulation of PD-L1 in vitro (Figure 3D). We speculated that there may be a targeting relationship between FOSL1 and PD-L1. We found two binding sites between FOSL1 and the PD-L1 promoter, namely, BS1 (−1420 to −1413 bp upstream of transcription start site (TSS)) and BS2 (−1864 to −1843 bp upstream of TSS). Their sequences are illustrated in Figure 3E. Subsequently, we found that FOSL1 only bound BS1 on the PD-L1 promoter and not BS2 (Figure 3F), as verified in Figure 3G. Compared to mutant BS1, FOSL1 was significantly bound to mutant BS2. These findings demonstrated that FOSL1 transcriptionally regulated PD-L1 levels in HSCC cells.

### 3.4. HOXA11-AS1 Enhanced FOSL1 mRNA Stability by Binding PTBP1

Since HOXA11-AS1 and FOSL1 both contributed to the expression of PD-L1, we focused on exploring their regulation. We firstly observed that FOSL1 levels decreased after silencing of HOXA11-AS1 (Figure 4A,B). To better investigate the function of HOXA11-AS1, we analyzed specific cellular fractions and found that HOXA11-AS1 was mainly distributed to the cytoplasm (Figure 4C), indicating that it may facilitate the expression of downstream genes at a post-transcriptional level. Therefore, we measured whether the FOSL1 mRNA stability was regulated by HOXA11-AS1. After actinomycin D treatment, we found that HOXA11-AS1 knockdown shortened the half-life of FOSL1 mRNA (Figure 4D), suggesting that HOXA11-AS1 stabilized it. Subsequently, Starbase predicted that both HOXA11-AS1 and FOSL1 could bind PTBP1, indicating that PTBP1 may be involved in the regulation of FOSL1 as a specific protein partner of HOXA11-AS1. RNA pulldown and RIP assays were carried out to confirm this speculation. As shown in Figure 4E,F, PTBP1 bound to the sense of HOXA11-AS1 and FOSL1 mRNA, and the abundance of PTBP1 binding to HOXA11-AS1 and FOSL1 mRNA was much higher than that of the IgG group, identifying that PTBP1 could act as a binding protein of HOXA11-AS1 and FOSL1. Furthermore, PTBP1 silencing induced the low expression of HOXA11-AS1 and FOSL1 in FaDu and Detroit 562 cells (Figure 4G), as well as shortening the half-life of HOXA11-AS1 and FOSL1 mRNA (Figure 4H–I). These findings indicated a targeting relationship between HOXA11-AS1 and PTBP1 and validated the enhancement effect of HOXA11-AS1 on the interaction between PTBP1 and FOSL1 mRNA.

### 3.5. HOXA11-AS1 Promoted PD-L1 Expression by Upregulating FOSL1 Levels through PTBP1, Thereby Facilitating Cell Immune Escape, Growth, and Metastasis

To further investigate the role of the HOXA11-AS1/PTBP1/FOSL1 axis on the regulation of PD-L1 and its biological function in vitro, we firstly stably overexpressed HOXA11-AS1 or PTBP1 by lentiviral transfection. The overexpression of HOXA11-AS1 or PTBP1 increased PTBP1, FOSL1, and PD-L1 levels in FaDu and Detroit 562 cells (Figure 5A,B). Then, shRNA against PTBP1 was used to silence PTBP levels. As expected, PTBP1 knockdown decreased the expression of PTBP1, FOSL1, and PD-L1, and more importantly, it partially reversed the upregulation of FOSL1 and PD-L1 caused by HOXA11-AS1 overexpression (Figure 5C). Interestingly, PD-L1 knockdown was subsequently used to exert the same reverse effects on the high expression of PD-L1 caused by HOXA11-AS1 overexpression, while the increased PTBP1 and FOSL1 levels showed no changes (Figure 5D). These data demonstrated that HOXA11-AS1 regulated PD-L1 by facilitating the association of PTBP1 with FOSL1. Since HOXA11-AS1 promoted PD-L1 levels, proliferation, metastasis, and immune escape in HSCC cells, we investigated the regulation mechanism of PD-L1 in HOXA11-AS1-overexpressed cells. As shown in Figure 5E, HOXA11-AS1 overexpression decreased CD8^+^ T cell percentages while increasing CD4^+^ T cell percentages. The knockdown of PD-L1 led to the opposite results and partially reversed the increase in T lymphocytes induced by HOXA11-AS1 overexpression. Moreover, PD-L1 knockdown increased the concentration of IFN-γ that was reduced by HOXA11-AS1 overexpression (Figure 5F), indicating that HOXA11-AS1 facilitated immune escape in HSCC cells by regulating PD-L1. Similarly, HOXA11-AS1 enhancement facilitated cell proliferation, migration, and invasion, while these effects were reversed by PD-L1 knockdown (Figure 5G–J). Therefore, we confirmed the role of the HOXA11-AS1/PD-L1/PTBP1/FOSL1 axis in promoting HSCC progression in vitro.

### 3.6. HOXA11-AS1 Knockdown Inhibited Immune Escape and Metastasis by Regulating PD-L1 and Downregulating FOSL1 In Vivo

To assess the anti-tumor effect of HOXA11-AS1 knockdown in vivo, PBMCs were firstly injected into NOD-SCID mice to establish a human immune system. Then, FaDu and Detroit 562 cells stably expressing shNC, shHOXA11-AS1-1, or shHOXA11-AS1-2 were subcutaneously injected into mice. We observed that, after silencing HOXA11-AS1 or treatment with PBMCs, the pathological degree and proliferation of the tumor and the expressions of PD-L1 and Ki-67 were reduced, while these effects were further enhanced by the knockdown of HOXA11-AS1 and the addition of PBMCs together (Figure 6A,B), indicating the effective anti-tumor role of HOXA11-AS1 deletion in vivo. The results suggested that tumors shrank after HOXA11-AS1 knockdown (Figure 6C), indicating that the inhibiting effect of T cells on tumor cells could be restored by HOXA11-AS1 knockdown.

Furthermore, we observed the decreased volume and weight of tumors after silencing HOXA11-AS1 (Figure 6D–F). Then, PTBP1, FOSL1, and PD-L1 levels decreased in vivo after HOXA11-AS1 knockdown (Figure 6G). Moreover, tumor metastasis capability was also affected by the depletion of HOXA11-AS1. As shown in Figure 6H–I, HOXA11-AS1 knockdown impaired the ability of the tumor to metastasize to the lung. Overall, the depletion of HOXA11-AS1 combined with PD-L1 blocking showed a more effective function in the inhibition of HSCC progression in vivo.

## 4. Discussion

One of the most difficult challenges during HSCC treatment is the invasion and metastasis to adjacent structures due to extensive lymphatic drainage [27,28]. Therefore, the investigation of new therapeutic targets has had a high profile in recent years. LncRNAs and their related mechanisms have also been extensively explored and are involved in many processes of cancer development, such as proliferation, differentiation, and metastasis [29,30,31,32]. Additionally, since PD-L1 is often reported to be overexpressed in malignant tumors and, mechanically, it helps cancer cells evade recognition by the host immune system, it has also become the focus of oncologic research [33,34]. Previous studies have reported the immunopathogenic effect and cancer-intrinsic function of PD-L1 in HSCC [13,35]; however, the contribution of PD-L1 to pathogenesis still needs further discussion overall. In this study, HOXA11-AS1 knockdown suppressed immune escape and metastasis in HSCC by downregulating PD-L1 levels and inhibiting the interaction between FOSL1 and PTBP1.

In recent years, the function of lncRNAs in HSCC has attracted increasing attention. For instance, lncRNA PEG10 was highly expressed in HSCC, and lncRNA PEG10 overexpression facilitated cell proliferation and metastasis in vitro [36]. HOXA11-AS was one of the most overexpressed lncRNAs in HSCC and was positively associated with lymph node metastasis [23]. We selected HOXA11-AS1 as our research object since it was upregulated in HSCC, as suggested by microarray analysis. The high expression of HOXA11-AS1 was measured both in tissues and cells. Functionally, HOXA11-AS1 silencing suppressed cell growth, metastasis, and the immune escape of HSCC cells to CD8^+^ T cells, which was a key inducement for tumor progression. Interestingly, we observed that PD-L1 expression positively correlated with HOXA11-AS1. The regulatory relationship between lncRNAs and PD-L1 had been reported in previous tumor-related studies [37,38]. For example, SNHG14 induced the inactivation of CD8^+^ T cells and the promotion of immune escape of diffuse large B cell lymphoma cells by activating PD-L1 [39]. This is consistent with our findings that the elimination of HOXA11-AS1 combined with PD-L1 antibody more effectively upregulated the content of CD8^+^ T cells in vitro, suggesting an association between HOXA11-AS1 and PD-L1 immune checkpoint for the first time in HSCC.

Exploring the biological targets related to HOXA11-AS1 or PD-L1 can better clarify the molecular mechanisms of HSCC and contribute to the development of HOXA11-AS1 clinical applications. Our findings indicated that FOSL1 bound the PD-L1 promoter and positively regulated its expression in vitro. FOSL1 was reported to act as an oncogene in a variety of cancers, and the tumor-intrinsic functions of FOSL1 have been widely investigated [40]. For instance, the FOSL1 transcription factor had been demonstrated to be a major effector of the RAS-ERK1/2 pathway and could increase the metastatic capability of lung cancer by activating epithelial–mesenchymal transition [41]. Our findings revealed other possible functions of FOSL1 in tumors, such as the promoting effect on the immune escape of tumor cells, by demonstrating an interaction between FOSL1 and PD-L1. PTBP1 is an RNA-binding protein that can shuttle from the nucleus to the cytoplasm, performing different functions mainly including mediating localization, translation initiation, and maintenance of mRNA stability [42,43]. In this study, HOXA11-AS1 was observed to be mostly distributed in the cytoplasm and bound to PTBP1, enhancing the stability of FOSL1 mRNA. This indicated that HOXA11-AS1 promoted the association of PTBP1 with FOSL1. Previous studies had suggested a similar function of other lncRNAs. Li et al. indicated that the lncRNA ANCR reduced the differentiation ability of human adipose-derived mesenchymal stem cells to definitive endoderm by enhancing the interaction between PTBP1 and ID2 [44]. Functionally, we found that PTBP1 silencing could reverse the upregulation effect of HOXA11-AS1 overexpression on FOSL1 and PD-L1. More importantly, PD-L1 knockdown blocked the inhibiting effect on CD8^+^ T cells and the promoting effect on HSCC cell growth and metastasis induced by HOXA11-AS1 overexpression. Therefore, we concluded that HOXA11-AS1 promoted the immune escape and metastasis of HSCC cells by upregulating PD-L1 and increasing the interaction between PTBP1 and FOSL1. To a certain extent, the role of this novel pathway had also been confirmed in vivo.

Due to the limitations of practical factors, in this study, the quantity of samples is not very high, and the applications of cell lines are not rich enough. We will conduct more comprehensive and accurate experiments in the future after more samples are collected and suitable new cell lines emerge. In any case, HOXA11-AS1 knockdown led to the decline of PD-L1 and inhibited the proliferation and metastasis of HSCC, which may also be affected by other factors, which is the key research to be explored in the future. 

## 5. Conclusions

Overall, our findings demonstrated a positive regulatory relationship between HOXA11-AS1 and PD-L1. HOXA11-AS1 knockdown suppressed PD-L1-mediated immune escape and metastasis by reducing the association between PTBP1 and FOSL1 in HSCC, providing a theoretical basis for HOXA11-AS1 as a potential prognostic marker for HSCC diagnosis and treatment.

## Figures and Tables

**Figure 1 cancers-14-03694-f001:** HOXA11-AS1 and PD-L1 were upregulated in HSCC, and HOXA11-AS1 positively correlated with PD-L1. (**A**,**B**) HOXA11-AS1 and PD-L1 expression in 40 HSCC tissues and matched peri-carcinomatous tissues measured by RT-qPCR. (**C**,**D**) Kaplan–Meier overall survival curve stratified by HOXA11-AS1 and PD-L1 expression. GraphPad was utilized to calculate the Kaplan–Meier plots, and 40 HSCC patients were included. The high/low expression of HOXA11-AS1 was defined by the median of its expression level, with higher than the median as high expression and lower than the median as low expression. Survival analysis was used. (**E**) The correlation between HOXA11-AS1 and PD-L1 expression was analyzed by Pearson’s correlation coefficient, *n* = 40. (**F**) PD-L1 expression in HSCC tissues was measured by immunohistochemical staining. (**G**,**H**) HOXA11-AS1 mRNA expression and PD-L1 protein levels in NP69 and HSCC cells (FaDu and Detroit 562 cells) detected by RT-qPCR and Western blot, respectively; *n* = 4. * *p* < 0.05, ** *p* < 0.01, *** *p* < 0.001. Original Blots see Appendix A.

**Figure 2 cancers-14-03694-f002:** HOXA11-AS1 knockdown suppressed PD-L1 expression and immune escape, proliferation, and metastasis of HSCC cells. FaDu and Detroit cells were transfected with two specific shRNAs, shHOXA11-AS1-1 and shHOXA11-AS1-2. (**A**) Transfection efficiencies were measured by RT-qPCR. (**B**,**C**) PD-L1 levels were detected after silencing HOXA11-AS1 by Western blot and immunofluorescence. Anti-PD-L1 antibodies restored the cytotoxic effect of T lymphocytes. Treated FaDu and Detroit cells were pretreated with or without anti-PD-L1 antibodies for 1 h and co-cultured with PBMCs for 72 h (Representative images (400×) are shown, bars = 100 µm.), then (**D**) the percentage of CD8^+^ and (**E**) CD4^+^ T cells was analyzed by flow cytometry, and (**F**) the concentration of IFN-γ was measured by enzyme-linked immunosorbent assay (ELISA). Viability, colony formation, migration, and invasion of FaDu and Detroit cells were measured by (**G**) CCK-8, (**H**) colony formation, (**I**) wound healing, and (**J**) transwell assays after knockdown of HOXA11-AS1. Representative images (200×) are shown. * *p* < 0.05, ** *p* < 0.01, *** *p* < 0.001. Original Blots see Appendix A.

**Figure 3 cancers-14-03694-f003:** FOSL1 was highly expressed in HSCC cells and positively regulated PD-L1 expression by binding the PD-L1 promoter. (**A**) FOSL1 expression in HSCC and peri-carcinomatous samples was tested by RT-qPCR. (**B**) Correlation between FOSL1 expression and HSCC patients’ survival rate. (**C**) FOSL1 protein levels in HSCC cells measured by Western blot. (**D**) FaDu and Detroit 562 cells were transfected with shFOSL1-1 and shFOSL1-2, and the expression of FOSL1 and PD-L1 was detected by Western blot. (**E**) JASPAR was used to predict the binding sites between FOSL1 and the promoter of PD-L1. (**F**,**G**) ChIP and dual-luciferase reporter assays were used to further validate the regulatory relationship between FOSL1 and the PD-L1 promoter. * *p* < 0.05, ** *p* < 0.01, *** *p* < 0.001. Original Blots see Appendix A.

**Figure 4 cancers-14-03694-f004:** HOXA11-AS1 enhanced FOSL1 mRNA stability by binding PTBP1. (**A**,**B**) FOSL1 levels were evaluated after HOXA11-AS1 knockdown using RT-qPCR and Western blot, respectively. (**C**) The cytoplasm and nucleus of FaDU cells were collected to measure the fraction of HOXA11-AS1. (**D**) FOSL1 mRNA decay in FaDu and Detroit cells measured after treating with actinomycin D (ACD) after HOXA11-AS1 knockdown. (**E**) Protein analysis of PTBP1 after RNA pulldown with biotin-labeled sense or antisense HOXA11-AS1 or FOSL1. (**F**) RIP assay was performed to measure the binding abundance of PTBP1 and FOSL1 mRNA or HOXA11-AS1. FaDu and Detroit cells were transfected with sh-PTBP1, then (**G**) PTBP1, FOSL1, and HOXA11-AS1 levels were evaluated by RT-qPCR, and (**H**) FOSL1 and (**I**) HOXA11-AS1 mRNA decay was measured after ACD treatment. * *p* < 0.05, ** *p* < 0.01, *** *p* < 0.001. Original Blots see Appendix A.

**Figure 5 cancers-14-03694-f005:** HOXA11-AS1 promoted PD-L1 expression by upregulating FOSL1 levels through PTBP1, thereby facilitating immune escape, growth, and metastasis of HSCC cells. FaDu and Detroit cells were transfected with pcDNA3.1-HOXA11-AS1 (HOXA11-AS1), pcDNA3.1-PTBP1 (PTBP1), sh-PTBP1, sh-PD-L1, or a combination of HOXA11-AS1 + sh-PTBP1 and HOXA11-AS1 + sh-PD-L1: (**A**) HOXA11-AS1, FOSL1, and PD-L1 levels were measured in FaDu and Detroit 562 cells treated with HOXA11-AS1 plasmid. (**B**) Relative expression of PTBP1, FOSL1, and PD-L1 was evaluated after overexpressing PTBP1. (**C**) Relative levels of PTBP1, FOSL1, and PD-L1 in HOXA11-AS1, shPTBP1, or HOXA11-AS1 + shPTBP1 groups. FaDu and Detroit cells were transfected with HOXA11-AS1, shPD-L1, or a combination of HOXA11-AS1 + shPD-L1: (**D**) Relative expression of PTBP1, FOSL1, and PD-L1 in cells transfected with HOXA11-AS1, sh-PD-L1, or HOXA11-AS1 + sh-PD-L1. Then, (**E**) the percentage of CD8^+^ and CD4^+^ T cells and (**F**) the concentration of IFN-γ were measured by flow cytometry and ELISA, respectively, and (**G**–**J**) the viability, colony formation, migration, and invasion of treated HSCC cells were analyzed by CCK-8, colony formation, wound healing, and transwell assays. Representative images (200×) are shown. * *p* < 0.05, ** *p* < 0.01, *** *p* < 0.001. Original Blots see Appendix A.

**Figure 6 cancers-14-03694-f006:** HOXA11-AS1 knockdown inhibited immune escape and metastasis by regulating PD-L1 and downregulating FOSL1 in vivo. FaDu and Detroit cells stably expressing shNC, shHOXA11-AS1-1, or shHOXA11-AS1-2 were injected subcutaneously into NOD-SCID mice to establish xenograft models. PBS or PBMCs were intraperitoneally injected into NOD-SCID mice to kill the xenografts more effectively. (**A**) HE, Ki67, and IHC staining were used to measure the tumor pathology, proliferation, and PD-L1 expression of xenografts treated with or without PBMCs or PBC. (**B**) The percentage of Ki67 cells was evaluated in xenografts. (**C**) The cytotoxic effect of treatment with or without PBMCs or PBS was measured by calculating xenograft volumes in shNC, shHOXA11-AS1-1, and shHOXA11-AS1-2 xenografts, and results are shown in the bar chart as tumor regression rate. (**D**–**F**) Tumor volume and weight were detected after HOXA11-AS1 knockdown in vivo. (**G**) RT-qPCR was used to detect HOXA11-AS levels, and Western blot was used to measure the expression of PD-L1, FOSL1, and PTBP1 after HOXA11-AS1 knockdown. (**H**,**I**) The ability of metastasis to the lung, tumor pathology, and nodule numbers of lung metastases were measured after of HOXA11-AS1 knockdown. * *p* < 0.05, ** *p* < 0.01, *** *p* < 0.001. Original Blots see Appendix A.

## Data Availability

All data generated or analyzed during this study are included in this published article.

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
