# Peer review of "HOXA11-AS1 Promotes PD-L1-Mediated Immune Escape and Metastasis of Hypopharyngeal Carcinoma by Facilitating PTBP1 and FOSL1 Association"

_cancers, 2022, doi:10.3390/cancers14153694_

Round 1

Reviewer 1 Report

The paper is interesting and suitable for publication, but some minor changes should be made before final acceptance:

1.- Important references have been omitted on the epidemiological background of head and neck cancer  (GLOBOCAN (IARC, WHO) update, new cases and deaths per year) and last evidences from the prognostic role of the PD1-PDL1 signaling pathway. The following papers should be referenced to update this paper:

Sung H, Ferlay J, Siegel RL, Laversanne M, Soerjomataram I, Jemal A, Bray F. Global Cancer Statistics 2020: GLOBOCAN Estimates of Incidence and Mortality Worldwide for 36 Cancers in 185 Countries. CA Cancer J Clin. 2021 May;71(3):209-249. doi: 10.3322/caac.21660. Epub 2021 Feb 4. PMID: 33538338.

Lenouvel D, González-Moles MÁ, Ruiz-Ávila I, Chamorro-Santos C, González-Ruiz L, González-Ruiz I, Ramos-García P. Clinicopathological and prognostic significance of PD-L1 in oral cancer: A preliminary retrospective immunohistochemistry study. Oral Dis. 2021 Mar;27(2):173-182. doi: 10.1111/odi.13509. Epub 2020 Jul 14. PMID: 32583572.

Birtalan E, Danos K, Gurbi B, Brauswetter D, Halasz J, Kalocsane Piurko V, Acs B, Antal B, Mihalyi R, Pato A, Fent Z, Polony G, Timar J, Tamas L. Expression of PD-L1 on Immune Cells Shows Better Prognosis in Laryngeal, Oropharygeal, and Hypopharyngeal Cancer. Appl Immunohistochem Mol Morphol. 2018 Aug;26(7):e79-e85. doi: 10.1097/PAI.0000000000000590. PMID: 29271789.

Hanna GJ, O'Neill A, Shin KY, Wong K, Jo VY, Quinn CT, Cutler JM, Flynn M, Lizotte PH, Annino DJ Jr, Goguen LA, Kass JI, Rettig EM, Sethi RKV, Lorch JH, Schoenfeld JD, Margalit DN, Tishler RB, Everett PC, Desai AM, Cavanaugh ME, Paweletz CP, Egloff AM, Uppaluri R, Haddad RI. Neoadjuvant and Adjuvant Nivolumab and Lirilumab in Patients with Recurrent, Resectable Squamous Cell Carcinoma of the Head and Neck. Clin Cancer Res. 2022 Feb 1;28(3):468-478. doi: 10.1158/1078-0432.CCR-21-2635. Epub 2021 Oct 19. PMID: 34667025.

2.- A better objective paragraph should be rewritten, avoiding the reporting of results in this paragraph, and transparently showing all the main components included in the study design (minimally, sample/study population, exposures/comparisons, outcomes, timing, design setting).

3.- This research has been approved by the Ethics Committee of the Third Xiangya Hospital of Central South University, but an IRB code has not been supplied. Please, submit.

4.- The authors consider the adjacent non-tumor tissue as a normal/healthy tissue control group: page 3, line 4 “…matched adjacent normal specimens were obtained from patients with HSCC…”. Accordingly, the methodology of this work is potentially biased, since the presence of genetically altered premalignant fields (with VERY important prognostic implications) is well established in patients with head and neck cancer. For this reason, the adjacent non-tumor tissue of these patients should not be used as a control group, and It should not be considered as a "healthy" sample. Therefore, the conclusions of the present work could be biased due to not having selected an adequate control group. Please remove this group, or change your interpretation, i.e., this group is not reflecting a healthy control group, but can be used to analyze the early steps of hypopharyngeal carcinogenesis. Of course, figures 1A and 1B should also be revised (i.e., normal vs tumour, missleading assumption).

5.- In relation with lab in vitro experiments, only three cell lines were cultured. This is an important limitation, due to the low repeatability of these experiments. Furthermore, it is not clear whether positive/negative controls were used in these experiments. Finally, neither in vitro nor in vivo animal experiments were blinded, a good practice seldom performed in these types of analyses.

6.- In relation to the statistical analysis, effect sizes should be reported for the survival analysis, preferably performing a multivariate cox regression analysis adjusted for potentially confounding variables. Only Kaplan-Meier curves were reported, but hazard ratios with 95% confidence intervals should be estimated, to account for the time-to-event nature of the variable  survival rate.

7.- As previously mentioned, a new paragraph should be included in the discussion section, critically showing the limitations of the present study. It is a mandatory paragraph in high impact factor journals. Authors could also address recommendations for future studies researching this topic.

Author Response

Reviewer 1

Comments and Suggestions for Authors

The paper is interesting and suitable for publication, but some minor changes should be made before final acceptance:

1.- Important references have been omitted on the epidemiological background of head and neck cancer (GLOBOCAN (IARC, WHO) update, new cases and deaths per year) and last evidences from the prognostic role of the PD1-PDL1 signaling pathway. The following papers should be referenced to update this paper:

Response: Thank you for your comments. We have added the latest number of cases and deaths of hypopharyngeal cancer in 2020 in the first paragraph of Introduction. Besides, the application of PD-1 in head and neck cancer is also discussed in the Introduction by referring to last literature. Please check the details in the manuscript.

Sung H, Ferlay J, Siegel RL, Laversanne M, Soerjomataram I, Jemal A, Bray F. Global Cancer Statistics 2020: GLOBOCAN Estimates of Incidence and Mortality Worldwide for 36 Cancers in 185 Countries. CA Cancer J Clin. 2021 May;71(3):209-249. doi: 10.3322/caac.21660. Epub 2021 Feb 4. PMID: 33538338.

Lenouvel D, González-Moles MÁ, Ruiz-Ávila I, Chamorro-Santos C, González-Ruiz L, González-Ruiz I, Ramos-García P. Clinicopathological and prognostic significance of PD-L1 in oral cancer: A preliminary retrospective immunohistochemistry study. Oral Dis. 2021 Mar;27(2):173-182. doi: 10.1111/odi.13509. Epub 2020 Jul 14. PMID: 32583572.

Birtalan E, Danos K, Gurbi B, Brauswetter D, Halasz J, Kalocsane Piurko V, Acs B, Antal B, Mihalyi R, Pato A, Fent Z, Polony G, Timar J, Tamas L. Expression of PD-L1 on Immune Cells Shows Better Prognosis in Laryngeal, Oropharygeal, and Hypopharyngeal Cancer. Appl Immunohistochem Mol Morphol. 2018 Aug;26(7):e79-e85. doi: 10.1097/PAI.0000000000000590. PMID: 29271789.

Hanna GJ, O'Neill A, Shin KY, Wong K, Jo VY, Quinn CT, Cutler JM, Flynn M, Lizotte PH, Annino DJ Jr, Goguen LA, Kass JI, Rettig EM, Sethi RKV, Lorch JH, Schoenfeld JD, Margalit DN, Tishler RB, Everett PC, Desai AM, Cavanaugh ME, Paweletz CP, Egloff AM, Uppaluri R, Haddad RI. Neoadjuvant and Adjuvant Nivolumab and Lirilumab in Patients with Recurrent, Resectable Squamous Cell Carcinoma of the Head and Neck. Clin Cancer Res. 2022 Feb 1;28(3):468-478. doi: 10.1158/1078-0432.CCR-21-2635. Epub 2021 Oct 19. PMID: 34667025.

2.- A better objective paragraph should be rewritten, avoiding the reporting of results in this paragraph, and transparently showing all the main components included in the study design (minimally, sample/study population, exposures/comparisons, outcomes, timing, design setting).

Response: Thank you for your comments. We have rewritten the last paragraph of Introduction. The new section introduces the sequence of our experiments, the general application method, and the significance of this study.

3.- This research has been approved by the Ethics Committee of the Third Xiangya Hospital of Central South University, but an IRB code has not been supplied. Please, submit.

Response: Thank you for pointing this out. We have added the IRB code (2020-S031) in the manuscript.

4.- The authors consider the adjacent non-tumor tissue as a normal/healthy tissue control group: page 3, line 4 “…matched adjacent normal specimens were obtained from patients with HSCC…”. Accordingly, the methodology of this work is potentially biased, since the presence of genetically altered premalignant fields (with VERY important prognostic implications) is well established in patients with head and neck cancer. For this reason, the adjacent non-tumor tissue of these patients should not be used as a control group, and It should not be considered as a "healthy" sample. Therefore, the conclusions of the present work could be biased due to not having selected an adequate control group. Please remove this group, or change your interpretation, i.e., this group is not reflecting a healthy control group, but can be used to analyze the early steps of hypopharyngeal carcinogenesis. Of course, figures 1A and 1B should also be revised (i.e., normal vs tumour, missleading assumption).

Response: Thank you for pointing this out, we are sorry for bringing such confusion in the manuscript, we have replaced the description of “normal tissue” with “peri-carcinomatous tissue” in the manuscript.

5.- In relation with lab in vitro experiments, only three cell lines were cultured. This is an important limitation, due to the low repeatability of these experiments. Furthermore, it is not clear whether positive/negative controls were used in these experiments. Finally, neither in vitro nor in vivo animal experiments were blinded, a good practice seldom performed in these types of analyses.

Response: Thank you for your comments. NP69 is the normal cell, while FaDu and Detroit562 are currently commercially available hypopharyngeal cancer cells. In view of the special location of hypopharyngeal cancer, there are few cell types of hypopharyngeal cancer available on the market at present. So, these two types of cells were selected for in vitro experiments.

6.- In relation to the statistical analysis, effect sizes should be reported for the survival analysis, preferably performing a multivariate cox regression analysis adjusted for potentially confounding variables. Only Kaplan-Meier curves were reported, but hazard ratios with 95% confidence intervals should be estimated, to account for the time-to-event nature of the variable survival rate.

Response: Thank you for your suggestions. We have analyzed the correlation between HOXA11-AS1 expression and the clinical characteristics of HSCC patients. As shown in Table 1, high HOXA11-AS1 expression was correlated with lymph node metastasis (p = 0.0245), distant metastasis (p = 0.0225), and TNM stages (p = 0.0230). Univariate and multivariate Cox regression analysis were also performed, and results indicated that HOXA11-AS1was an independent factor affecting HSCC patients’ survival (Table 2 and 3). Univariate analysis: HR, 3.92; 95% CI, 2.01–6.47, p = 0.0053; multivariate analysis: HR, 2.76; 95% CI, 1.86-5.27, p = 0.0122. Please check details in Table 1, 2, and 3 in the manuscript.

7.- As previously mentioned, a new paragraph should be included in the discussion section, critically showing the limitations of the present study. It is a mandatory paragraph in high impact factor journals. Authors could also address recommendations for future studies researching this topic.

Response: Thank you for your suggestions. We have added a new paragraph to describe the limitations of this work. Please check the details in the manuscript.

Reviewer 2 Report

The manuscript entitled “HOXA11-AS1 promotes PD-L1–mediated immune escape and metastasis of hypopharyngeal carcinoma by facilitating PTBP1 and FOSL1 association” which was contributed by Zhou et al. provided a novel regulation network among PD-L1, long non-coding RNA HOXA11-AS1 and PTBP1 and FOSL1 in hypopharyngeal squamous cell carcinoma (HSCC). As we known, PD-L1 is critical protein in cancer immune escaping. Its regulation mechanism will be important in HNSC tumorigenesis. However, the reason why the authors focus on HOSA11-AS1 and PTBP1 in HNCC were quite weak in the introduction section, the authors should increase their background information and importance in cancer or in HSCC. Following are some comments for the authors that should be revised in their manuscript

1.      The authors need to improve the background information of HOSA11-AS1 and PTBP1 in HNCC or other cancer in the introduction section.

2.      The IRB and lab-animal permission number should be included in the material and methods section.

3.      The GAPDH primer sequence is missing in the RT-qPCR section.

4.      PDL1 is the surface receptor, the authors should reveal the exact surface PD-L1 expression level by flow cytometry or confocal microscope.

5.      The flow cytometry results should be represented by original histogram or dot accumulation graph.

6.      The sample size and 95% CI should be included in the K-M plotter.

7.      The authors need to explain that how did they normalize the protein expression such as FOSL1, PD-L1 expression in their quantification results.

8.      The expression level of HOSA11-AS1 and PTBP1 should be included in figure 5.

9.      Please explain that why the HOSA11-AS1 regulated the in vivo tumor immune escape in the immunodeficient mice Nod-SCID mice since they do have mature T cells?

Author Response

Reviewer 2

Comments and Suggestions for Authors

Zhou et al. show in their article that HOXA11-AS1 promotes PD-L1–mediated immune escape and metastasis of hypopharyngeal squamous cell carcinoma (HSCC) by facilitating PTBP1 and FOSL1 association.

The authors use a variety of assays to address the role of lncRNA HOXA11-AS1 in HSCC. However, there are numerous issues that still need to be addressed:

- HOXA11-AS1 knockdown results in decreased tumor cell proliferation. So it is not clear whether (to what extent) the effects of HOXA11-AS1 are due to decreased tumor cell proliferation vs. PD-L1-mediated immune escape.

Response: Thank you for your suggestions. In Figure 2, knockdown of HOXA11-AS1 can reduce cell proliferation, invasion and migration; At the same time, knocking down HOXA11-AS1 can also cause the decrease of CD3+/CD4+T cells, while the increase of CD3+/CD8+T cells and the secretion of IFN-γ, which indicates that HOXA11-AS1 can cause immune escape. The tumor cells escape the killing of immune cells, which in turn promotes cell proliferation and metastasis to some extent. Please refer to PMID: 34796777. Of course, there are other mechanisms by which HOXA11-AS1 affects other target proteins to regulate tumor cell proliferation through PTBP1, which is also where we need to focus on later research, please refer to PMID:34446576.

- The interpretation of some results are not supported by the assays used (e.g. immune escape; cytotoxic effect of T lymphocytes; see detailed comments below). This needs to be corrected.

- Several paragraphs in the methods lack important details. Some are listed exemplarily below.

The issues below need to be carefully addressed:

- Page 3: The paragraph about patients and IHC is lacking essential details (e.g. were this biopsies or surgical specimens? What were the patient characteristics and tumor stage? Samples for IHC: were these FFPE samples? Which secondary antibodies were used, what concentration, incubation time? …)

Response: Thank you for pointing this out, we have added the detail information of patients, including the criteria for inclusion and exclusion criteria.

In IHC experiments, we have added more details to describe the procedure. Besides, the concentration and No. of secondary antibody used in Western blot, IHC and Immunofluorescence assays were supplemented. 

- Fig. 1C+D: Labeling of the x-axis is missing

Response: Thank you for pointing this out, we have added the label of the x-axis (month) of Fig. 1C and 1D.

- Fig. 1C+D: Which database was used to calculate the Kaplan-Meier plots? How many patients were included in each group? How was high/low HOXA11-AS1 defined? What statistical test was used? All this information should be mentioned in the figure legend.

Response: Thank you for pointing this out, please refer to the details below. All information has also been added in the figure legend of our manuscript.

GraphPad was utilized to calculate the Kaplan-Meier plots, and forty HSCC patients were included. The high/low expression of HOXA11-AS1 was defined by the median of its expression level, with higher than the median as high expression and lower than the median as low expression. Survival analysis was used.

- Fig. 1E: I assume that PD-L1 and the lncRNA were measured in tumor tissue of the HSCC patients? How many patients; n=40? Please state this in the figure legend.

Response: Thank you for your suggestions, the number of the HSCC patients is 40, we have added this information in the figure legend.

- Page 7, chapter 3.1, last sentence: for clarity reasons, the sentence should read “… in HSCC cell lines Detroit 562 and FaDu, compared to NP69, a normal human nasopharyngeal epithelial cell line.”

Response: Thank you for your suggestions, we have revised the sentence into “Moreover, we detected increased expression of HOXA11-AS1 and PD-L1 in HSCC cell lines Detroit 562 and FaDu, compared to NP69, a normal human nasopharyngeal epithelial cell line.”

- How many independent experiments were performed in Fig. 1G and 1H? Please add n=.

Response: Thank you for pointing this out, the n value is 4, we have added it in the figure legend.

- Figure 2 needs to be re-arranged. Figure 2D and 2E should be next to each other, as they show the same thing just with another cell line.

Response: Thank you for your suggestions, we have put Figure 2D and 2E together.

- Figure 2D and 2E: These figures do not show that “anti-PD-L1 antibodies restored the cytotoxic effect of T lymphocytes”. The data simply show the percentage of total CD8 and total CD4 cells (percentage of what? CD45+? CD45+ and tumor cells?)

Response: Thank you for your question, we evaluated the percentage of CD3+/CD4+ and CD3+/CD8+ T cells in our study. Please refer to PMID: 31747941.

- Page 4, 2.10: The co-culture experiment needs to be described in much more detail. How many cancer cells and how many PBMC were used, which plates were used, which medium, what was the concentration of the anti-PD-L1 antibodies, from which company, which clone? What was the absolute number of the cancer cells and PBMCs at the end of the experiment (after 72 hours)?

Response: Thank you for your suggestions, we have added more details below. Information has also included in the Materials and Methods part of the manuscript.

FaDu and Detroit 562 cells with or without PD-L1 antibody (2 µg/mL, Abcam) treatment were selected for co-culture experiments. Human peripheral blood mononuclear cells (PBMCs) were isolated by Ficoll-Hypaque gradient centrifugation. For co-culture, FaDu and Detroit 562 cells were plated into 24-well plates in Modified Eagle Medium (DMEM) with a density of 1×105 cells/well, PBMCs were placed in culture insert (0.4 µm pore size, Corning, USA) in DMEM at a concentration of 4×105 cells/well.

- Figure 2G-J: Each of these figures need to be described more clearly. Which subfigure describes viability, which one the CCK-8 assay, how many experiments were performed.

Response: Thank you for your suggestions, we have revised this in the manuscript.

- Page 7, paragraph 3.2: what about a direct measurement of CD8 T cell-mediated tumor cell killing (analysis of tumor cell death by CD8 T cells using co-culture and 7-AAD/CFSE labeling) or indirect (expression of granzymes / perforin in CD8 T cells by FACS).

Response: Thank you for your suggestions, we tested the percentage of CD3+/CD4+ and CD3+/CD8+ T cells in our study. Considering the time limit, we will do this in the follow-up study.

- Figure 3A: Is FosL1 also highly expressed in HSCC patient samples, as shown in Fig. 1F for PD-L1? Does high/low FosL1 expression in HSCC patients affect survival? IHC for FosL1 in HSCC patient samples and Kaplan-Meier plots for FosL1 should be added.

Response: Thank you for your suggestions, we have tested the expression of FOSL1 in HSCC patients, we have also analyzed the correlation between FOSL1 expression and HSCC patients’ survival rate. Using RT-qPCR, we found that FOSL1 was significantly elevated in HSCC tumor samples (Figure 3A). Moreover, we found that HSCC patients with high expression of FOSL1 exhibited a worse survival rate than those with low FOSL1 expression (Figure 3B).

- Figure 5D: Same as for Fig. 2D, the data simply show the percentage of total CD8 and total CD4 cells. With this analysis alone, one cannot conclude immune escape or increased tumor cell-specific killing by CD8 T cells.

Response: Thank you for your question, we evaluated the percentage of CD3+/CD4+ and CD3+/CD8+ T cells in our study. Please refer to PMID: 31747941.

- Figure 5H: Why does sh-PD-L1 reduce tumor growth in the absence of immune cells?

Response: Sorry for our negligence, we have added an appropriate amount of PBMCs to HSCC cells, which has been revised in the Methods. See the manuscript for details.

- Figure 6E-F: FACS analysis of the tumor-immune-microenvironment should have been performed to demonstrate that PD-L1 upregulation promotes immune escape (altered number of CD8 T cells or CD8 T cell function/cytotoxicity)

Response: Thank you for your suggestion. We did not test for CD8T cells, considering that nude mice do not have T cells. Knocking down HOXA11-AS1 leads to the decline of PD-L1, which may also be affected by other factors, which is the key to be explored in the future.

- Figure 6H: The quality of the tumor pictures is too low in the PDF file to clearly see all the metastatic nodules.

Response: Thank you for pointing this out. And we have provided higher resolution of tumor images.

Reviewer 3 Report

Zhou et al. show in their article that HOXA11-AS1 promotes PD-L1–mediated immune escape and metastasis of hypopharyngeal squamous cell carcinoma (HSCC) by facilitating PTBP1 and FOSL1 association.

The authors use a variety of assays to address the role of lncRNA HOXA11-AS1 in HSCC. However, there are numerous issues that still need to be addressed:

-          HOXA11-AS1 knockdown results in decreased tumor cell proliferation. So it is not clear whether (to what extent) the effects of HOXA11-AS1 are due to decreased tumor cell proliferation vs. PD-L1-mediated immune escape.

-          The interpretation of some results are not supported by the assays used (e.g. immune escape; cytotoxic effect of T lymphocytes; see detailed comments below). This needs to be corrected.

-          Several paragraphs in the methods lack important details. Some are listed exemplarily below.

The issues below need to be carefully addressed:

-          Page 3: The paragraph about patients and IHC is lacking essential details (e.g. were this biopsies or surgical specimens? What were the patient characteristics and tumor stage? Samples for IHC: were these FFPE samples? Which secondary antibodies were used, what concentration, incubation time? …)

-          Fig. 1C+D: Labeling of the x-axis is missing

-          Fig. 1C+D: Which database was used to calculate the Kaplan-Meier plots? How many patients were included in each group? How was high/low HOXA11-AS1 defined? What statistical test was used? All this information should be mentioned in the figure legend.

-          Fig. 1E: I assume that PD-L1 and the lncRNA were measured in tumor tissue of the HSCC patients? How many patients; n=40? Please state this in the figure legend.

-          Page 7, chapter 3.1, last sentence: for clarity reasons, the sentence should read “… in HSCC cell lines Detroit 562 and FaDu, compared to NP69, a normal human nasopharyngeal epithelial cell line.”

-          How many independent experiments were performed in Fig. 1G and 1H? Please add n=.

-          Figure 2 needs to be re-arranged. Figure 2D and 2E should be next to each other, as they show the same thing just with another cell line.

-          Figure 2D and 2E: These figures do not show that “anti-PD-L1 antibodies restored the cytotoxic effect of T lymphocytes”. The data simply show the percentage of total CD8 and total CD4 cells (percentage of what? CD45+? CD45+ and tumor cells?)

-          Page 4, 2.10: The co-culture experiment needs to be described in much more detail. How many cancer cells and how many PBMC were used, which plates were used, which medium, what was the concentration of the anti-PD-L1 antibodies, from which company, which clone? What was the absolute number of the cancer cells and PBMCs at the end of the experiment (after 72 hours)?

-          Figure 2G-J: Each of these figures need to be described more clearly. Which subfigure describes viability, which one the CCK-8 assay, how many experiments were performed

-          Page 7, paragraph 3.2: what about a direct measurement of CD8 T cell-mediated tumor cell killing (analysis of tumor cell death by CD8 T cells using co-culture and 7-AAD/CFSE labeling) or indirect (expression of granzymes / perforin in CD8 T cells by FACS)

-          Figure 3A: Is FosL1 also highly expressed in HSCC patient samples, as shown in Fig. 1F for PD-L1? Does high/low FosL1 expression in HSCC patients affect survival? IHC for FosL1 in HSCC patient samples and Kaplan-Meier plots for FosL1 should be added.

-          Figure 5D: Same as for Fig. 2D, the data simply show the percentage of total CD8 and total CD4 cells. With this analysis alone, one cannot conclude immune escape or increased tumor cell-specific killing by CD8 T cells.

-          Figure 5H: Why does sh-PD-L1 reduce tumor growth in the absence of immune cells?

-          Figure 6E-F: FACS analysis of the tumor-immune-microenvironment should have been performed to demonstrate that PD-L1 upregulation promotes immune escape (altered number of CD8 T cells or CD8 T cell function/cytotoxicity)

-          Figure 6H: The quality of the tumor pictures is too low in the PDF file to clearly see all the metastatic nodules.

Author Response

Reviewer 3

Comments and Suggestions for Authors

The manuscript entitled “HOXA11-AS1 promotes PD-L1–mediated immune escape and metastasis of hypopharyngeal carcinoma by facilitating PTBP1 and FOSL1 association” which was contributed by Zhou et al. provided a novel regulation network among PD-L1, long non-coding RNA HOXA11-AS1 and PTBP1 and FOSL1 in hypopharyngeal squamous cell carcinoma (HSCC). As we known, PD-L1 is critical protein in cancer immune escaping. Its regulation mechanism will be important in HNSC tumorigenesis. However, the reason why the authors focus on HOSA11-AS1 and PTBP1 in HNCC were quite weak in the introduction section, the authors should increase their background information and importance in cancer or in HSCC. Following are some comments for the authors that should be revised in their manuscript

  1. The authors need to improve the background information of HOXA11-AS1 and PTBP1 in HNCC or other cancer in the introduction section.

Response: Thank you for pointing this out. We have added more background information of HOXA11-AS1 and PTBP1 in the introduction part of our manuscript.

  1. The IRB and lab-animal permission number should be included in the material and methods section.

Response: Thank you for pointing this out, the IRB code (2020-S031) has been provided.

  1. The GAPDH primer sequence is missing in the RT-qPCR section.

Response: Thank you for pointing this out, however, we have provided GAPDH primer sequence in the RT-qPCR section in original manuscript.

  1. PDL1 is the surface receptor, the authors should reveal the exact surface PD-L1 expression level by flow cytometry or confocal microscope.

Response: Thank you for your suggestion, we tested the expression of PD-L1 using immunofluorescence and signals were observed under confocal microscope (Figure 2C).

  1. The flow cytometry results should be represented by original histogram or dot accumulation graph.

Response: Thank you for your suggestion, however, we can’t put flow cytometry images in our manuscript due to typographical problems.

  1. The sample size and 95% CI should be included in the K-M plotter.

Response: Thank you for your suggestion, the related information have been provided in Table 1-3.

  1. The authors need to explain that how did they normalize the protein expression such as FOSL1, PD-L1 expression in their quantification results.

Response: Thank you for your suggestion, and we normalized the expression of FOSL1 and PD-L1 through comparing the internal reference protein (GAPDH or β-actin). Please see the Methods in the manuscript.

  1. The expression level of HOSA11-AS1 and PTBP1 should be included in figure 5.

Response: Thank you for your suggestions. We have added the related expression of HOXA11-AS1 and PTBP1 in Figure 5A, 5B, 5C and 5D.

  1. Please explain that why the HOSA11-AS1 regulated the in vivo tumor immune escape in the immunodeficient mice Nod-SCID mice since they do have mature T cells?

Response: Thank you for your question, we injected human immunocyte mixtures containing PBMCs (4 × 106/per mouse) and activated T cells (1 × 106/per mouse) or PBS in to the mice when the tumor volume was more than 100 mm3 by tail intravenous injection once a week for three times. Information has also added in the in vivo xenograft experiments part of the manuscript.

Round 2

Reviewer 2 Report

The primer sequence is miss again in "5ʹ- -3ʹ (antisense) for FOSL1". Please fix it. 

Author Response

Reviewer 2

Comments and Suggestions for Authors

The primer sequence is miss again in "5ʹ- -3ʹ (antisense) for FOSL1". Please fix it. 

Response: Thank you for your suggestions. We’ve add the primer sequences of FOSL1 to the manuscript in the part of Methods Page 3 Paragraph 3.
